# Quantifying Paddling Kinematics through Muscle Activation and Whole Body Coordination during Maximal Sprints of Different Durations on a Kayak Ergometer: A Pilot Study

**DOI:** 10.3390/ijerph20032430

**Published:** 2023-01-30

**Authors:** Y. M. Garnier, P. M. Hilt, C. Sirandre, Y. Ballay, R. Lepers, C. Paizis

**Affiliations:** 1EA3920 Prognostic Factors and Regulatory Factors of Cardiac and Vascular Pathologies, University of Franche-Comté, 25000 Besançon, France; 2INSERM 1093-CAPS, UFR des Sciences du Sport, Université Bourgogne, 21000 Dijon, France; 3Centre for Performance Expertise, Faculté des Sciences du Sport, BP 27 877, Université de Bourgogne, 21078 Dijon, France

**Keywords:** flatwater sprint kayaking, body kinetics, surface electromyography, pacing strategy, ergometry

## Abstract

Paddling technique and stroke kinematics are important performance factors in flatwater sprint kayaking and entail significant energetic demands and a high strength from the muscles of the trunk and upper limbs. The various distances completed (from 200 m to 1000 m) require the athletes to optimize their pacing strategy, to maximize power output distribution throughout the race. This study aimed to characterize paddling technique and stroke kinematics during two maximal sprints of different duration. Nine nationally-trained participants (2 females, age: 18 ± 3 years; BMI: 22.2 ± 2.0 Kg m^−1^) performed 40 s and 4 min sprints at maximal intensity on a kayak ergometer. The main findings demonstrated a significantly greater mean stroke power (237 ± 80 W vs. 170 ± 48 W; *p* < 0.013) and rate (131 ± 8 spm vs. 109 ± 7 spm; *p* < 0.001) during the 40 s sprint compared to the 4 min sprint. Athletes used an all-out strategy for the 40 s exercise and a parabolic-shape strategy during the 4 min exercise. Despite the different strategies implemented and the higher muscular activation during the 40 s sprint, no change in paddling technique and body coordination occurred during the sprints. The findings of the present study suggest that the athletes constructed a well-defined profile that was not affected by fatigue, despite a decrease in power output during the all-out strategy. In addition, they regulated their paddling kinematics during the longer exercises, with no change in paddling technique and body coordination.

## 1. Introduction

Flatwater sprint kayaking events encompass races competed over 200 m, 500 m, and 1000 m in single- or crew-boats composed of 2 or 4 athletes. In single kayaking, the world’s best race times range from 33 s to 37 s over 200-m, to 200 s and 230 s over 1000 m for men and women, respectively (data from the International Canoe Federation), which makes aerobic and anaerobic capacity, as well as high upper body strength, strong physiological factors in performance [1,2,3]. Furthermore, the cyclic nature of the activity also makes paddling technique an important performance factor that athletes should optimize to ensure the effectiveness of each blade stroke to propel the boat, despite the persistent unbalanced state resulting from the thinness of the boat.

From a global perspective, Lopez-Lopez and Ribas Serna identified a similar paddling technique among Olympic athletes, who qualified for an “optimal stroke profile” [4]. Qualitatively, Limonta and colleagues reported longer stroke lengths being developed by elite athletes compared to national-level athletes at the end of an incremental 4 min test [5]. Over a 200 m sprint, Pickett and colleagues pointed out that among elite athletes, those of higher performance levels also performed longer stroke lengths in 200 m races than elite athletes of a lower performance level [6,7]. International elite athletes of a higher performance level were therefore distinguished from national level athletes by a greater range of motion in their knee extension, which increased their pelvis and trunk rotation amplitude and, ultimately, increased the stroke length through a well-forward catching phase [5,8]. However, they maintain the exit phase once the blade reaches the vertical position, to quickly initiate the contralateral catching phase, while reducing the ineffective portion of the stroke [7]. In particular, Gomes and colleagues reported that increasing the stroke rate permitted a more rectangular force profile to be produced from the blade–water interaction, which results in a greater impulse during the stroke and a higher mean boat velocity [7]. Accordingly, the increase in mean stroke rate during 200 m races has paralleled improvements in performances over the years [9], which identifies stroke rate as a performance factor separating international elite athletes and athletes of a national level [6].

The contribution of the legs in flatwater kayaking should also not be discarded. Providing the feet are strapped onto the footrest, kayakers perform push and pull actions with their legs to enhance their pelvis and trunk rotation [5,8,10]. Nilsson and Rosdahl estimated the duration of the synchronized period between leg extension and paddle thrust at 0.2 s [10]. Although of short duration, the synchronization between the leg extension, pelvis, and trunk rotation, as well as paddle thrust, needs to be finely refined, to provide a gain in boat velocity quantified from 6% [11] to 16% [10].

It should, however, be mentioned that the vast majority of studies described paddling technique for a reduced number of stroke cycles and with no consideration of the changes that could occur throughout a race. The decrease in boat speed during flatwater sprint kayaking races is now well-established, even in elite Olympian athletes [12]. Only a few studies have analyze stroke kinematics over the time course of 200 m or 500 m maximal sprints [6,13,14]. Over short-distance sprints (i.e., 200-m), a reduction in boat speed and stroke rate was identified in young national-level paddlers [14] and international and national-level athletes [6]. Vaquero-Cristóbal and colleagues also identified a reduction in stroke efficiency over the time course of a 200 m sprint [14]. During a 500 m sprint, Bertozzi and colleagues also observed reductions in stroke length and velocity, while stroke duration remained constant [13]. These elements shed light on the occurrence of fatigue that athletes face during a race, affecting their performance. Given the different distances competed in flatwater kayaking (i.e., from 200 m to 1000 m), athletes employ different pacing strategies to optimize power distribution throughout the sprint and to limit the occurrence of fatigue [15,16]. Therefore, pacing strategies represent a performance factor in flatwater sprint kayaking [17]. These strategies also impact stroke peak power, power distribution or stroke rate, and physiological responses during sprinting [17,18]. For instance, male athletes increase stroke rate and reduce stroke length to produce a greater stroke impulse during paddling. Despite the lack of studies regarding sprints of longer distances (e.g., 1000 m), one could suggest that the distribution of stroke rate or power during a 1000 m sprint race depends on the pacing strategy used. Consequently, athletes would also likely adapt their paddling technique (paddle displacement and body coordination) differently depending on the distance covered and their pacing strategy.

Adaptations to stroke kinematics and paddling technique can result from body coordination changes. Bjerkefors and colleagues identified, for instance, that elite athletes increased their shoulder flexion and adduction, and performed trunk flexion and rotation of greater amplitude, to increase power output during paddling [19]. In addition, Bertozzi and colleagues evidenced that elite athletes increased their lateral trunk flexion over a 500 m maximal sprint, to compensate for the fatigue-induced reduction in the amplitude of shoulder range of movement [13]. However, they did not detect any changes in knee or ankle range of motion during the sprint [13]. In addition to motion capture, surface electromyography (sEMG) recordings can assist the investigation of body coordination related to paddling technique and evidence changes in the activation pattern of the muscles with the occurrence of fatigue. To date, the pattern of muscle recruitment during kayaking has, however, been scarcely investigated and limited to comparisons of shoulder muscle recruitment between on-water kayaking and with an ergometer [20] or for the lower limbs in with-water races [21]. In this context, it remains unknown how athletes adapt their paddling technique during a prolonged sprint when facing fatigue.

Therefore, this study aimed to identify whether national-level athletes adapted different paddling techniques and stroke kinematics during simulated maximal sprints of different durations. Additionally, we intended to identify how the different pacing strategy employed by athletes impacts the occurrence of fatigue and its effects on body coordination and muscle activation patterns. We hypothesized that athletes would use a more conservative pacing strategy during the 4 min sprint compared to an all-out strategy employed during the 40 sec sprint, limiting the kinematic and muscle coordination changes.

## 2. Materials and Methods

### 2.1. Participants

Nine experienced young kayak athletes (2 females, age: 18 ± 3 years; BMI: 22.2 ± 2.0 Kg.m^−1^; training experience: 8.0 ± 5.2 years) participated in this study. All the athletes provided their consent to participate in the study. All competed at least at national level, and two were members of the U18 French kayak team. The athletes trained on average 10 h to 14 h per week, including on-water kayaking, strength training, and running. The study conformed to the standard set by the World Medical Association Declaration of Helsinki “Ethical Principles for Medical Research Involving Human Subjects” (2008). 

### 2.2. Experimental Design

Participants visited the laboratory in a single session to perform two maximal efforts on a flywheel kayak ergometer (Dansprint PRO kayak ergometer, Dansprint ApS, Hvidovre Municipality, Denmark). All participants were experienced with paddling on an ergometer for their personal training. The distance between the seat and the foot bar was set for each athlete to correspond with the settings of their kayak. The flywheel resistance was adjusted to account for the athlete’s weight to reproduce on-water drag forces, following the manufacturer’s prescription (http://www.dansprint.com) (accessed on 27 December 2022). Participants were instructed not to undertake vigorous training 24 h before the experiment.

Before the experimental measurements, participants performed a warm-up, including 6 min paddling at an intensity of 70% of their maximal perceived effort, 2 min at 90% of their maximal perceived effort, and three 10 s maximal efforts. Recovery periods of 2 min followed the 6 min and the 2 min sets, with 1 min rest between the 10 s efforts. Then, participants performed two maximal exercises of different durations of 40 s and 4 min, to correspond with average durations for 200 m and 1000 m, with 15 min of recovery in between. Athletes were asked to achieve the highest distance they could during each bout, with no instructions about pacing strategy and no feedback on their performance. 

### 2.3. Stroke Kinematics

Stroke rate and power were recorded stroke-by-stroke and stored for off-line analysis using the ergometer software (Dansprint analyser V1.12, Dansprint ApS, Hvidovre Municipality, Denmark). Data were averaged over 15 complete cycles, defined by the position of the paddle tip [22], and analyzed at three-time points, corresponding to the beginning (T1), middle (T2), and end (T3) of each exercise.

### 2.4. Surface Electromyography (sEMG)

EMG activity was recorded during exercises using wireless EMG probes (Cometa systems, Milan, Italy) at a frequency of 1000 Hz. After shaving and cleaning the skin with alcohol, pairs of pre-gelled surface electrodes (10 mm diameter) were applied to the midpoint of the palpated muscle belly along the muscle fibers with a 2 cm center-to-center interval. Ten upper and lower limb muscles were investigated: anterior, middle, and posterior deltoids (*Del*); upper trapezius (*Trap*); pectoralis major (*Pec*); latissimus dorsi (*Lat*); biceps brachii (*Bic*); triceps brachii (*Tri*); rectus abdominis (*Rect*); and vastus lateralis (*Vast*). The activity of the three portions of the deltoids was averaged to reflect the activity of the global deltoids. EMG activity was averaged over 1 stroke cycle and analyzed with automatic processing (Matlab, Mathworks, Torrance, CA, USA).

### 2.5. Motion Capture

Three-dimensional marker trajectory was recorded using seven infra-red cameras sampling at 200 Hz (Vicon T-series, Vicon motion systems Ltd., Oxford, UK). Nineteen retro-reflective markers (diameter = 20 mm) were applied on the body to identify the trunk, upper- (forearm, arm, shoulder girdle) and lower-limb (pelvis, thigh, leg), and the paddle (see Figure 1). Based on the 3D movement of these markers, we defined an elevation angle for each segment, as the angles between each segment and the vertical axis (defined by gravity-extrinsic reference). In addition, the amplitude of the paddle displacement in the vertical and horizontal plans (X, Z) was also evaluated. To assess the coordination between the different segments during the movement, a principal component analysis (PCA) was performed on the time series of the 10 elevation angles (upper-arm, arm, trunk, leg, and lower-leg for each hemibody). A correlation matrix was used to normalize our data set and to consider the different ranges of motion of each segment. To account for different motor strategies, separate PCAs were performed for each participant and each condition (for more details on the computation, see [23,24]). Here we used, as a coordination value, the percentage associated with the three first principal components for 15 double-stroke cycles, at the beginning, middle, and end of each test (40 s vs. 4 min). 

### 2.6. Statistical Analysis

All data are presented as mean ± standard deviation (SD) in the text, figures, and tables. The nature of the distribution was assessed for all variables using a Shapiro–Wilk test, to ensure the proper use of parametric ANOVA. A Greenhouse–Geisser correction to the degree of freedom was applied when the sphericity of the data was violated. A two-way 2 × 3 ANOVA was used to test the effect of condition (40 sec vs. 4 min exercise) and time (T1 vs. T2 vs. T3) on EMG activity, kinematics, and stroke rate and power. When significant, the main effect and condition × time interactions were followed up with a Tuckey HSD test. Effects sizes are reported as partial eta squared (*ηp*^2^) and Cohen’s *dz*, the latter being calculated from the mean and standard deviation of the variables, and the correlation between these variables using G*Power (version 3.1, Universität, Düsseldorf, Germany). Statistical analysis was performed with Statistica (StatSoft France, version 7.1, STATISTICA). The significance level was set at 0.05 (two-tailed) for all analyses.

## 3. Results

### 3.1. Stroke Rate and Power

The condition × time interaction for the stroke rate (*p* = 0.007; *ηp*^2^ = 0.464) revealed a significantly lower stroke rate during the 4 min exercise compared to the 40 s at each time point (all *p* < 0.001; all *dz* > 0.921; see Figure 2A). Stroke rate was significantly lower at T2 compared to T1 and T3 during the 4 min exercise (all *p* < 0.018; all *dz* > 1.528), with no difference between T1 and T3. Stroke rate remained constant during the 40 s exercise (all *p* > 0.747; all *dz* < 0.680). The ANOVA also detected a condition × time interaction for stroke power (*p* < 0.001; *ηp*^2^ = 0.598; see Figure 2B). Power was greater during the 40 s exercise compared to the 4 min at each time point (all *p* < 0.013; all *dz* > 0.508) and significantly higher at T1 than T2 and T3 in each condition (all *p* < 0.031; all *dz* > 1.148). Power increased from T2 to T3 during the 4 min exercise (*p* = 0.005; *dz* = 1.521) but remained constant for the 40 s exercise (*p* = 0.971; *dz* = 0.265).

### 3.2. Electromyographic Activity

Figure 3 presents the EMG activity of each muscle at the different time points during the 40 s and the 4 min exercises. Due to a technical issue during recording, the EMG data of one participant were removed from analyses, and only the data from eight participants were considered. A significant main effect of time was detected for *Pec*, *Trap,* and *Lat* muscles (all *p* < 0.006; all *ηp*^2^ > 0.516), with no condition or condition × time interactions (all *p* > 0.057; all *ηp*^2^ < 0.434). All three muscles demonstrated greater EMG activities at T2 compared to T3 (all *p* < 0.004; all *dz* > 1.336). EMG activity was also greater at T2 compared to T1 for the *Trap* muscle (*p* < 0.001; *dz* = 1.675).

A significant condition × time interaction was detected for the *Bic*, *Tri*, *Del*, *Rect,* and *Vast* muscles (all *p* < 0.049; all *ηp*^2^ > 0.350). The *Rect* muscle showed a significantly greater EMG activity at T1 and T3 during the 40 s than during the 4 min exercise (all *p* < 0.026; all *dz* > 0.809). All other muscles (i.e., *Bic*, *Tri*, *Del,* and *Vast*), showed a greater EMG activity during the 40 s than the 4 min exercise at each time point (all *p* < 0.001; all *dz* > 0.873). The *Vast* muscle showed no change of the EMG activity over time during both exercises (all *p* > 0.062; all *dz* < 0.917). During the 4 min exercise, the *Tri* and *Del* muscles showed greater EMG activities at T2 compared to T1 and T3 (all *p* < 0.018; all *dz* > 1.218), and the EMG activity of the *Bic* was greater at T2 compared to T3 (*p* = 0.021; *dz* > 0.764). No other changes were observed.

### 3.3. Kinematic Data

Table 1 shows the mean elevation angle of the upper arm, arm, and trunk calculated for the two conditions. Due to technical problems with reflective markers, only complete data from four subjects were included in the kinematic analysis. The ANOVA revealed neither main effects of condition or time (all *p > 0.271;* all *Bic < 0.353*) nor interactions (*p =* 0.279; *ηp*^2^ = 0.364) for elevation angle coordination. 

## 4. Discussion

This study aimed to describe the kinematics and muscle activity of the paddling technique in young, experienced paddlers during maximal sprints of different durations on a kayak ergometer, as well as to analyze how athletes adapt their pacing strategy depending on race duration. In line with our hypothesis, the results of stroke power highlighted that the athletes used a more conservative pacing strategy during the 4 min sprint than in an the all-out strategy employed during the 40 s sprint, with higher muscular activity in the biceps brachii, triceps, deltoids, rectus femoris, and vastus lateralis muscles during the 40 s sprint. Despite these differences, no changes were found in the kinematics and muscle coordination.

The mean power (237 ± 80 W and 170 ± 48 W) and stroke rate (131 ± 8 spm and 109 ± 7 spm) recorded during the 40 s and 4 min sprint, respectively, were similar to those reported for national-level athletes by [6] over 200 m and somewhat lower than those reported for international athletes by [19]. Maximal and mean power and stroke rate were significantly lower during the 4 min bout compared to the 40 s sprint, despite the same resistance being imposed by the ergometer. During the 40 s sprint, the stroke rate remained constant, while the stroke power decreased immediately after the start (i.e., from T1 to T2 and T3), with no further decrease between T2 and T3. During the 4 min sprint, the stroke rate and power were higher at T1 compared to T2 and T3, and both parameters increased from T2 to T3. These elements showed that athletes implemented two different pacing strategies during exercise. Athletes implemented an all-out strategy during the 40 s sprint, characterized by a greater power being developed at the start and then maintained as long as possible, while an even-pace strategy, characterized by a parabolic-shape profile, was implemented during the 4 min exercise [16,17]. These two strategies, previously reported for well-trained kayakers [6,17] and rowers [25], may improve performance through optimization of the balance between power production and the occurrence of fatigue. Specifically, an all-out strategy would enable athletes to achieve a greater amount of work above the critical power before a high level of fatigue limited their performance [15]. On the contrary, athletes may implement an even-pace strategy based on their own experience, to limit disturbance in the muscle intracellular milieu [17], therefore delaying the occurrence of fatigue [25] and allowing them to limit the decrease in boat speed or even increase it. 

Performance in sprint flatwater kayaking also relies upon the acquisition of a fine paddling technique that ensures the effectiveness of the paddling stroke throughout the race. In line with the “optimal stroke profile” identified by Lopez-Lopez and Ribas Serna as shared by international elite athletes [4], the present findings showed the same paddling profile being implemented by the athletes during both sprints (see Figure 4). Furthermore, no change in this pattern was evidenced during the race, despite different pacing strategies. Previously, well-trained and elite athletes also demonstrated a synchronized action between the upper and the lower limbs [10,19]. Our findings showed that the elevation angles and body coordination were similar between the 40 s and the 4 min sprint and were similar between the left and right sides of the body, as reported by Bjerkefors and colleagues [19]. Hence in accordance with Therrien, Colloud, and Begon [18], the present findings suggest that well-trained athletes kept the same paddling technique, despite having different stroke rates between sprints. 

Surface EMG was also recorded in our study, to characterize the impact of different pacing strategies and the occurrence of fatigue on muscle activation patterns. Muscle activities of the upper (i.e., Bic, Tri, Del) and lower limbs (i.e., Vast) were consistently higher during the 40 s compared to the 4 min sprint. The greater mean activity of these muscles during the 40 s sprint corroborates the higher power output recorded. Precisely, the increase in mean EMG activity of the biceps brachii represented a greater level of force being developed to resist against elbow extension during the pulling phase (biceps brachii). In contrast, an increase in triceps brachii activity limits elbow flexion of the aerial limbs, to ensure the effectiveness of the opposite pulling phase [19,26]. The greater activity of the vastus lateralis muscle recorded during the 40 s sprint reflects the prominent role the lower limbs play in force transmission in flatwater kayaking [10,11]. However, most of the investigated trunk muscles (i.e., Trap, Pec, Lat) did not demonstrate a different EMG activity between the sprint modalities, except for a greater activity for the rectus abdominis at T1 and T3 during the 40 s compared to the 4 min sprint. The trapezius and the latissimus dorsi muscles control shoulder extension and internal rotation during kayak paddling [20]. The similar magnitude of EMG activity between the sprint modalities for these two muscles suggests that the greater power output produced during the 40 s exercise did not increase shoulder instability. Together, these findings suggest that athletes adapt to the greater power output mainly by increasing muscular force to ensure the effectiveness of their paddling technique, rather than changing their paddling tip kinematics (e.g., pulling stroke length).

Flatwater race sprint kayaking performance is a complex process that depends on numerous factors, such as athletes’ level, inter-individual history, balance, and distance. Using a kayak ergometer coupled with motion capture represents an alternative strategy for characterizing the paddling stroke profile of individual athletes. Studies investigating changes in paddling stroke profile during maximal sprints could help coaches to more precisely identify athletes’ weaknesses, in order to build a well-suited paddling technique, develop an alternative pacing strategy, and improve on-water and strength-training programs. It would therefore be of interest to further explore in real conditions the muscular activations associated with kinematics and body coordination with a larger sample size. Such studies may also provide support for identifying the stroke impulse distribution and kinematics of different athletes and help coaches in determining optimal athlete positioning in crew boats to optimize the stroke synchronization between athletes. Kayak ergometers have been shown to replicate well the physiological demands of on-water kayaking during maximal testing [27], allowing prescription and monitoring of training intensities [28]. However, differences in stroke profile, and particularly in shoulder and arm range of motion and muscles activations, were reported during paddling on a kayak ergometer compared to on-water kayaking [26,29,30]. These elements, therefore, necessitate interpreting the paddling stroke profile obtained on a kayak ergometer with caution when implementing these findings on-water, to avoid any detrimental effects that would impair stroke efficiency and performance or induce overuse-related injuries. The findings of the present study agree with previous studies, where athletes demonstrated similar paddling stroke profiles. However, the limited number of participants in the present study prevents the identification of consistent changes in body coordination related to the occurrence of fatigue during maximal sprint exercise. Future studies should focus on body kinematics, to identify precisely how athletes adapt their paddling technique to limit the impact of neuromuscular fatigue on stroke efficiency.

## 5. Conclusions

The main findings of this study show that athletes used different pacing strategies between a 40 s and a 4 min maximal sprint. Specifically, athletes implemented an all-out strategy during the 40 s sprint, to achieve a greater amount of work above the critical power before the occurrence of neuromuscular fatigue, while employing an even-paced strategy to avoid excessive accumulation of fatigue early on during the race. Despite these two markedly different strategies, athletes kept a similar paddling technique, which could be defined as the “optimal stroke profile” reported in previous studies. Only an increase in the mean activity of the upper and lower limb muscles was detected when comparing the 40 s and 4 min sprints, which could be related to an increase in force to resist the greater stroke power and maintain the effectiveness of their paddling technique. Therefore, we encourage practitioners to develop the most suitable strategy relative to their racing distance and give more attention to strength training of the biceps brachii, triceps brachii, deltoids, rectus femoris, and vastus lateralis, which were further activated to produce more power from a similar paddling technique and body coordination.

## Figures and Tables

**Figure 1 ijerph-20-02430-f001:**
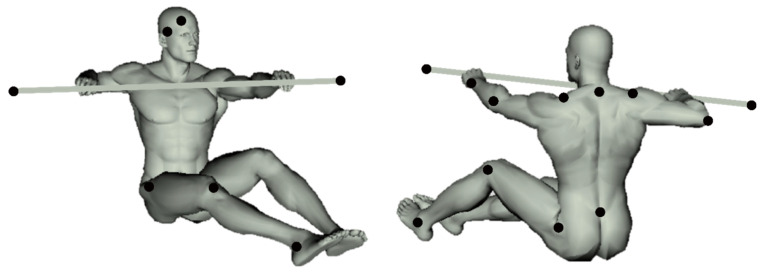
Placement of the retro-reflective markers on the participants.

**Figure 2 ijerph-20-02430-f002:**
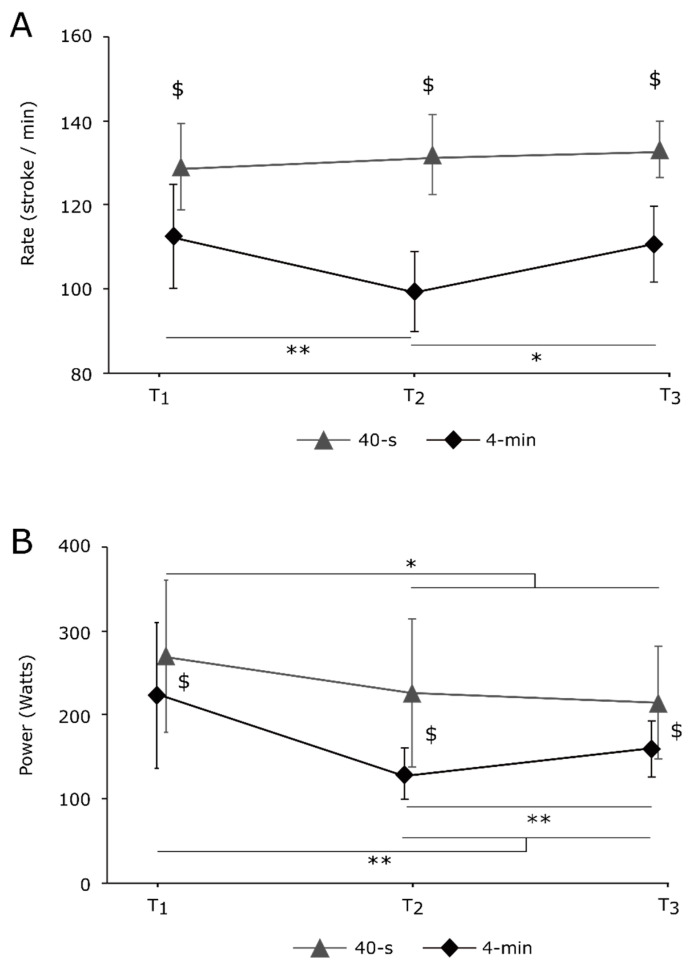
Stroke rate (**A**) and power (**B**) recorded during the 40 s and 4 min sprints (N = 9; mean ± SD). * denotes a significant difference between time points within condition, and ^$^ difference between condition. One item (*p* < 0.05) and two items (*p* < 0.01).

**Figure 3 ijerph-20-02430-f003:**
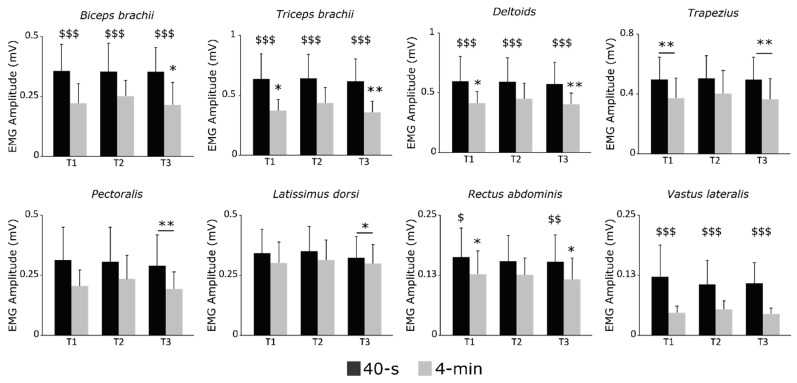
EMG amplitude of muscle activations recorded at the beginning (T1), middle (T2), and end (T3) of each sprint (N = 8; mean ± SD). * denotes a significant difference with T2, and ^$^ difference between condition at the corresponding time point. One item (*p* < 0.05), two items (*p* < 0.01) and three items (*p* < 0.001).

**Figure 4 ijerph-20-02430-f004:**
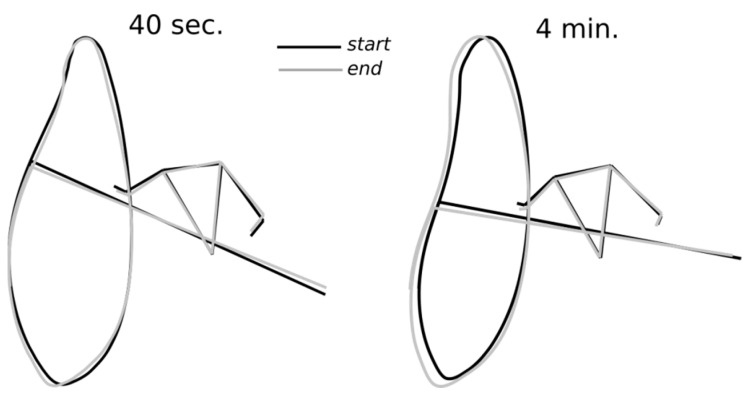
Mean start posture and paddle tip trajectory for the 15 first cycles (black) and 15 last cycles (grey) of the 40 s (**left**) and 4 min (**right**) condition, for a typical subject.

**Table 1 ijerph-20-02430-t001:** Mean and standard deviation across subjects of the main kinematic parameters for the 40 s and 4 min conditions.

	40 s.		4 min.	
	Start	End	Start	End
**Right Hemibody**				
Elev. Angle upper arm (deg.)	67 ± 2	64 ± 2	67 ± 3	65 ± 2
Elev. Angle arm (deg.)	70 ± 4	72 ± 3	68 ± 3	68 ± 2
Elev. Angle trunk (deg.)	14 ± 2	14 ± 1	14 ± 1	14 ± 1
Amplitude Paddle Vert. (cm)	164 ± 15	167 ± 15	160 ± 14	162 ± 7
Amplitude Paddle Hrzt. (cm)	61 ± 9	63 ± 8	64 ± 11	66 ± 12
**Left Hemibody**				
Elev. Angle upper arm (deg.)	67 ± 2	64 ± 2	67 ± 3	65 ± 2
Elev. Angle arm (deg.)	70 ± 4	72 ± 3	68 ± 3	68 ± 2
Elev. Angle trunk (deg.)	14 ± 2	14 ± 1	14 ± 1	14 ± 1
Amplitude Paddle Vert. (cm)	158 ± 2	154 ± 3	158 ± 2	153 ± 7
Amplitude Paddle Hrzt. (cm)	67 ± 12	71 ± 11	74 ± 7	74 ± 5
Coordination (%)	94 ± 2	94 ± 2	94 ± 2	93 ± 3

## Data Availability

Individual data of the present study are confidential, as extracted from the athletes’ monitoring program.

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
