# Peer review of "Quantifying Paddling Kinematics through Muscle Activation and Whole Body Coordination during Maximal Sprints of Different Durations on a Kayak Ergometer: A Pilot Study"

_ijerph, 2023, doi:10.3390/ijerph20032430_

Round 1
Reviewer 1 Report
The general idea of the study is good. There are some aspects to improve the quality of the investigation as the following:
1) The 3 distances in flatwater kayaking are 200, 500 and 1000m. Please, correct it.
2) The introduction is very long and the explanations of some aspects are not necessary such as: lines 48 to 60 and 100 to 110 aprox
3) There are some interesting studies to take into consideration and key for result interpretation here:
Alberty, M., Sidney, M., Pelayo, P., & Toussaint, H. M. (2009). Stroking characteristics during time to exhaustion tests. Medicine & Science in Sports & Exercise, 41(3), 637–644. https://doi.org/10. 1249/MSS.0b013e31818acfba
Borges, T. O., Bullock, N., & Coutts, A. J. (2013). Pacing characteristics of international Sprint Kayak athletes. International Journal of Performance Analysis in Sport, 13(2), 353–364. https:// doi.org/10.1080/24748668.2013.11868653
Abellán-Aynés, O. López-Plaza,D. Martínez-Aranda, L.M. & Alacid, F. (2022): Inter-stroke steadiness: a new kinematic variable related to 200m performance in young canoeists, Sports Biomechanics, DOI: 10.1080/14763141.2022.2071327
4) Why the sample were not exclusively men and women?
5) Software and other equipment must be defined with the name of the company and the country (i.e. line 146)
6) The procedures of electrodes placement for sEMG should have been doing according to standards such as the ones described by Cram et al., 1998
7) The design and the presentation of Figure 2 and table 1 are confusing and the quality is questionable. Please, improve it.
8) References style must be reviewed
9) Include limitations of the study such as the limited number of participants
10) English grammar and expressions must improve. A deep review by a native speaker is needed.
Author Response
The general idea of the study is good. There are some aspects to improve the quality of the investigation as the following:
We thank the editor and the reviewer for their time devoted to our manuscript and we hope that our responses and amendments will satisfy your concerns.
1) The 3 distances in flatwater kayaking are 200, 500 and 1000m. Please, correct it.
Sentence rephrase to avoid misinterpretation (see line 32)
2) The introduction is very long and the explanations of some aspects are not necessary such as: lines 48 to 60 and 100 to 110 aprox
According to the reviewer’s concern, we deleted explanations that we believe were not primarily of importance and rephrased accordingly the section from lines 48 to 60. However, we firmly believe that the information reported from lines 100 to 110 are of primary importance to provide a relevant understanding of paddling kinematics and its analysis performed in previous studies
3) There are some interesting studies to take into consideration and key for result interpretation here:
Alberty, M., Sidney, M., Pelayo, P., & Toussaint, H. M. (2009). Stroking characteristics during time to exhaustion tests. Medicine & Science in Sports & Exercise, 41(3), 637–644. https://doi.org/10. 1249/MSS.0b013e31818acfba
Borges, T. O., Bullock, N., & Coutts, A. J. (2013). Pacing characteristics of international Sprint Kayak athletes. International Journal of Performance Analysis in Sport, 13(2), 353–364. https:// doi.org/10.1080/24748668.2013.11868653
Abellán-Aynés, O. López-Plaza,D. Martínez-Aranda, L.M. & Alacid, F. (2022): Inter-stroke steadiness: a new kinematic variable related to 200m performance in young canoeists, Sports Biomechanics, DOI: 10.1080/14763141.2022.2071327
We thank the reviewer for providing these references. With respect to the topic of the present study, we added the study from Borges et al., 2013 (see line 78), but dismiss the studies from Alberty 2009 and Abellan-Aynés 2022, that we believe could not reliably support our statements.
4) Why the sample were not exclusively men and women?
We were not aware of different pacing strategies or paddling techniques between men and women that would led us to question the inclusion of men or women only. Furthermore, this study was conducted in the context of the performance monitoring of the athlete from the local training center so that all kayak athletes were included in this protocol.
5) Software and other equipment must be defined with the name of the company and the country (i.e. line 146)
We thank the reviewer for raising this point. Completed as requested (see line 148).
6) The procedures of electrodes placement for sEMG should have been doing according to standards such as the ones described by Cram et al., 1998
Details about skin shaving and cleaning with alcohol were added in the text (see line 154) as recommended by the SENIAM procedure (Hermes et al., 2000).
7) The design and the presentation of Figure 2 and Table 1 are confusing and the quality is questionable. Please, improve it.
We corrected table 1 to remove typing errors and enhance the presentation of the table. As for figure 1, figure 2 has been exported with a resolution of 600 PPP which we believe is adapted for the present purpose. Signs, marks and color lines used in figure 2 are different between sprint duration, and statistical differences have been highlighted similarly to what is usually done in many other articles. We would be grateful if the reviewer could precise which elements need to be enhanced.
8) References style must be reviewed
Amended as required
9) Include limitations of the study such as the limited number of participants
We added a limitation section as requested (see lines 325-330).
10) English grammar and expressions must improve. A deep review by a native speaker is needed.
An English review has been conducted accordingly.
Reviewer 2 Report
Dear Researchers
Please kindly consider the following comments and provide necessary revisions and specify point-by-point responses:
1- your research title is : "Quantifying paddling kinematic through muscle activation and whole body coordination during maximal sprint of different 3 duration on a kayak ergometer: a pilot study"
Whats the possible application of such research in real situation? is it merely a fundamental study?
2- include pvalues in result section of abstract.
3- check key words based on Mesh standard.
4-include demographic for participants in methodde section of abstract.
5- include appropriate references for first paragraph of introduction. (especially the last line of it).
6- since your research theme is regarding the biomechanics, better to start the introduction with the focus of biomechanics (its importance on sports performance) science and its importance. you can use the following reference.
Didehdar, Daryoush, and Ameneh Kharazinejad. "The Association Between Sprint Speed Test and Isokinetic Knee Strength in Healthy Male Volleyball Players." International Journal of Sport Studies for Health 4.2 (2021).
Rusdiana, Agus. "3D Kinematics Analysis of Overhead Backhand and Forehand Smash Techniques in Badminton." Annals of Applied Sport Science 9.3 (2021): 0-0.
7- the training protocol should be referenced in methode section.
8- result section is excellent.
9- strength and weakness of study should be reported in discussion section. then, put study limitation.
10- recommendation for future studies should be stated at the end of discussion.
Author Response
Dear Researchers
Please kindly consider the following comments and provide necessary revisions and specify point-by-point responses:
We thank the reviewer for his/ her time devoted to review our manuscript and we hope that our responses and amendments will satisfy your concerns.
1- your research title is: "Quantifying paddling kinematic through muscle activation and whole-body coordination during maximal sprint of different 3 duration on a kayak ergometer: a pilot study"
Whats the possible application of such research in real situation? is it merely a fundamental study?
This study sought to detect changes in stroking kinematics occurring during maximal sprints when athletes faced fatigue. In a training context, the findings of this study may help to individualize the training content (on water and strength training) and paddling technique of each athlete. In addition, providing a reliable identification of the stroke impulse during the pulling phase, these data can help optimize athletes’ position in crew boats to enhance synchronization of power distribution between athletes. These elements have been added as perspectives at the end of the discussion section (see lines 302-310).
2- include pvalues in result section of abstract.
Done as requested
3- check key words based on Mesh standard.
We check and amend keywords according to Mesh standard (flatwater sprint kayaking, body kinetics, surface electromyography) and add “ergometry”
4-include demographic for participants in methodde section of abstract.
Added as requested (see line 19-20)
5- include appropriate references for first paragraph of introduction. (especially the last line of it).
According to the reviewer’s concern, we add the reference from Abbis and Laursen (2008) to support the pacing strategy suggested in this sentence. However, to our knowledge no published study supports the assumption about the unbalance state when kayaking on water, this observation being made from our experience in sprint kayaking.
6- since your research theme is regarding the biomechanics, better to start the introduction with the focus of biomechanics (its importance on sports performance) science and its importance. you can use the following reference.
Didehdar, Daryoush, and Ameneh Kharazinejad. "The Association Between Sprint Speed Test and Isokinetic Knee Strength in Healthy Male Volleyball Players." International Journal of Sport Studies for Health 4.2 (2021).
Rusdiana, Agus. "3D Kinematics Analysis of Overhead Backhand and Forehand Smash Techniques in Badminton." Annals of Applied Sport Science 9.3 (2021): 0-0.
We thank the reviewer for his/ her valuable comment. However, our focus is not primarily related to biomechanics. We believe that focusing on performance to start the introduction better reflects how we want this study to be considered.
7- the training protocol should be referenced in methode section.
We provided the average training volume of the participants and the main activities performed during training in the methods section (see lines 124-125). Precisions about the exact training schedule and program of the athletes would make this section long and out of the primary context of this study.
8- result section is excellent.
We thank the reviewer for his / her encouraging comment.
9- strength and weakness of study should be reported in discussion section. then, put study limitation.
We add strengths and limitations of the present study as requested (see lines 308-327)
10- recommendation for future studies should be stated at the end of discussion.
We provided recommendations for future studies regarding the necessity to identify coordination between body segments while fatigue occurs during maximal sprints (see lines 332-336).
Reviewer 3 Report
Its a good and interesting article.
In my opinion, the PCA for measuring coordination should be briefly explained.
- The format of table 1 needs to be reviewed
Author Response
its a good and interesting article.
Thank you for your positive comment. We hope that the amendments provided about the PCA analysis would satisfy your concern.
In my opinion, the PCA for measuring coordination should be briefly explained.
Further details about the PCA analysis have been added in the motion capture section (see lines 171 – 180).
The format of table 1 needs to be reviewed
Revised accordingly